# Association of liver fibrosis biomarkers with overall and CVD mortality in the Korean population: The Dong-gu study

**Seong-Woo Choi**[1], **Sun-Seog Kweon**[2], **Young-Hoon Lee**[3], **So-Yeon Ryu**[1], **Hae-Sung Nam**[4], **Min-Ho Shin**[2]*

**1** Department of Preventive Medicine, Chosun University Medical School, Gwangju, Republic of Korea,
**2** Department of Preventive Medicine, Chonnam National University Medical School, Hwasun, Jeollanam-do, Republic of Korea, **3** Department of Preventive Medicine & Institute of Wonkwang Medical Science, Wonkwang University School of Medicine, Iksan, Jeollabuk-do, Republic of Korea, **4** Department of Preventive Medicine, Chungnam National University Medical School, Daejeon, Republic of Korea

* mhshinx@naver.com

## Abstract

This study evaluated the associations of liver fibrosis biomarkers [non-alcoholic fatty liver disease fibrosis score (NFS), fibrosis-4 (FIB-4), aspartate aminotransferase/platelet ratio index (APRI), and BARD score] with mortality in Korean adults aged ≥50 years. We analyzed 7,702 subjects who participated in Dong-gu Study. The associations of liber fibrosis biomarkers with mortality were investigated using Cox proportional hazards models. Overall mortality increased with increasing NFS level [adjusted hazard ratio (aHR) 4.3, 95% confidence interval (CI) 3.3–5.5 for high risk vs. low risk], increasing FIB-4 level (aHR 3.5, 95% CI 2.9–4.4 for high risk vs. low risk), and increasing APRI level (aHR 3.5, 95% CI 2.1–5.8 for high risk vs. low risk) but not with BARD score. The Harrell's concordance index for overall mortality for the NFS and FIB-4 was greater than that for the APRI and BARD score. In conclusion, NFS, FIB-4, and APRI showed a significant relationship with the overall mortality, and NFS and FIB-4 showed a significant relationship with the CVD mortality after adjustment for covariates. In addition, the NFS and FIB-4 were more predictive of overall mortality than the APRI and BARD score in Korean adults aged ≥50 years.

## Introduction

As the global obesity rate rises, nonalcoholic fatty liver disease (NAFLD) has emerged as one of the most common chronic liver diseases (CLDs). The global prevalence of NAFLD is approximately 25.2% [1], with Korea having a 32.9% prevalence due to a westernized lifestyle [2]. NAFLD is a common disease, but some cases are dangerous because nonalcoholic steatohepatitis, or liver fibrosis, can progress to cirrhosis or liver cancer. Thus far, liver biopsy has been the gold standard for diagnosing liver fibrosis in patients with NAFLD. However, because liver biopsies are invasive, the patient faces the risk of complications and high costs. As a result, new biomarkers have been developed, including the NAFLD fibrosis score

**Data Availability Statement:** Some restrictions by the Institutional Review Board of Chonnam National University Hospital will apply. Due to ethical or legal issues (data contain potentially

identifying or sensitive patient information), data can be accessed on request through IRB deliberations (such as a list of collaborators). Interested researchers may contact data manager of the Dong-gu study Chang Kyun Choi (gin4567@paran.com, Chonnam National University Medical School) to request data access.

**Funding:** The authors received no specific funding for this work.

**Competing interests:** The authors have declared that no competing interests exist.

(NFS), fibrosis-4 (FIB-4), aspartate aminotransferase (AST)/platelet ratio index (APRI), and BARD score [3].

The previous meta-analysis, which included several cohort studies, examined the relationship between biomarkers and liver fibrosis in patients with NAFLD [4–6] and whether biomarkers could predict long-term outcomes in patients with NAFLD [7–10]. As a result, the European Association for the Study of the Liver (EASL) and the American Association for the Study of Liver Diseases (AASLD) guidelines endorsed the NFS and FIB-4 as a diagnostic measure for ruling out advanced liver fibrosis [11,12].

However, data on the general population that included asymptomatic patients with CLD were insufficient, and studies with the Korean general population have been limited. Since the leading causes of burden of disease in Korea have been liver cirrhosis and liver cancer and the leading cause of death among Koreans in their 40s and 50s has been liver diseases [13], it is necessary to investigate the liver fibrosis biomarkers with the Korean population.

The objectives of this study were to 1) assess the associations of NFS, FIB-4, APRI, and BARD score with overall and cardiovascular disease (CVD) mortality and 2) compare the performances of NFS, FIB-4, APRI, and BARD score using the Harrell's concordance index (c-index) [14], adaptation of area under of receiver operating characteristic curve in a general Korean population of 50 years.

## Materials and methods

### Subjects

The Dong-gu study is a prospective study implemented to assess risk factors of chronic diseases among Korean citizens aged ≥50 years. Details of the Dong-gu study were described in a previous publication [15]. Potential participants were identified using Korean national resident registration records. Trained researchers called 34,040 residents aged 50 and up in the Dong-gu district of Gwangju Metropolitan City, South Korea. Finally, from 2007 to 2010, 9,260 people participated in the baseline study. There were 1,558 subjects excluded from this study for the following reasons: no blood test (n = 134), history of liver diseases such as cirrhosis, cancer, or hepatitis (n = 339), no alcohol data (n = 134), or significant alcohol intake (30 g/day in men and 20 g/day in women) (n = 951). In total, 7,702 people were included in the study. This study was conducted in accordance with the Helsinki Declaration guidelines, and all participants provided informed consent. The institutional review board of Chonnam National University Hospital approved the study protocol (No. I-2008-05-056).

### Data collection

All subjects were interviewed by trained examiners using a questionnaire designed to elicit information on the status of current smoking, alcohol intake, regular walking, and the diagnosis of hypertension (HTN) and diabetes. Height and weight were measured to the nearest 0.1 cm and 0.1 kg, respectively. Body weight (in kg) divided by height squared (in m$^2$) yielded the body mass index (BMI). After subjects had rested for at least 5 min in a seated position, blood pressure was measured on the right upper arm with a mercury sphygmomanometer (Baumanometer; WA Baum Co., Inc., Copiague, NY, USA) and an appropriately sized cuff. The first appearance (phase I) and disappearance (phase V) of Korotkoff's sounds were used to calculate systolic blood pressure (SBP) and diastolic blood pressure (DBP), which was then read to the nearest 2 mmHg. Three consecutive SBP and DBP measurements were obtained, and the average values were used in the analysis.

Following 12-h overnight fasts, blood samples were drawn from antecubital veins; serum samples were separated within 30 min and then stored at −70°C before analysis. Using enzymatic

methods on an automatic analyzer (Hitachi-7600; Hitachi, Ltd., Tokyo, Japan), the concentrations of AST, alanine aminotransferase (ALT), gamma-glutamyl transpeptidase (GGT), total cholesterol, triglycerides, high-density lipoprotein (HDL) cholesterol, platelet count, and serum albumin were measured. Glycated hemoglobin ($HbA_{1c}$) levels were assessed by high-performance liquid chromatography using the VARIANT II system (Bio-Rad, Hercules, CA, USA).

### Ascertainment of death

Linking with National Statistical Office registries yielded death information. The date of death was ascertained until December 31, 2020. The International Classification of Diseases, 10th revision (ICD-10) was used to code the causes of death. Cardiovascular deaths were classified as ICD-10 codes I20–25.

### Measurement of liver fibrosis

Liver fibrosis was assessed by four biomarkers: NFS, FIB-4, APRI, and BARD score.

NFS = −1.675 + 0.037 × age (years) + 0.094 × BMI ($kg/m^2$) + 1.13 × impaired fasting glucose or diabetes (yes = 1, no = 0) + 0.99 × AST/ALT ratio −0.013 × platelet count ($10^9$/L) − 0.66 × serum albumin (g/dL) [16].

FIB-4 = age (years) × AST (U/L) / platelet count ($10^9$/L) × ALT $(U/L)^{1/2}$ [17].

APRI = 100 × AST (U/L)/(upper limit of normal)/platelet count ($10^9$/L), where the upper limit of normal is 34 U/L for females and 36 U/L for males [18].

BARD score = BMI ≥28 $kg/m^2$ = 1 point; AST/ALT ratio ≥0.8 = 2 points; and diabetes mellitus (DM) = 1 point [19].

Based on previous studies [16–19], low/intermediate/high risk categories for each biomarker were provided; for NFS, <−1.455, −1.455–0.676, and >0.676; for FIB-4, <1.30, 1.30–2.67, and >2.67; for APRI, <0.5, 0.5–1.5, and >1.5; and for BARD score, 0–1, 2–3, and 4.

### Statistical analysis

All values are given as a percentage or as a mean standard deviation. The $\chi^2$ or Student's t-test was used to compare differences between now-deceased and deceased subjects. To approximate normal distributions, waist circumference, SBP, $HbA_{1c}$, total cholesterol, triglycerides, HDL cholesterol, and GGT data were log-transformed. Analysis of variance was used to evaluate variable comparisons based on biomarker categories. Cox proportional hazards models were used to examine the relationships between liver fibrosis biomarkers and overall and CVD mortality. The initial model was unadjusted. The second model was adjusted for sex, log-waist circumference, smoking, alcohol intake, regular walking, hypertension, log-SBP, log-$HbA_{1c}$, log-total cholesterol, log-triglycerides, log-HDL, and log-GGT. Diabetes for FIB-4, age and diabetes for APRI, and age for BARD score were additionally included and adjusted in the second model. Kaplan-Meier curves of cumulative overall mortality, compared by log-rank tests, were used for assessing different incidence of death between individuals with low, intermediate and high risk of NFS, FIB-4, APRI, and BARD score.

In addition, using the *somersd* package, the Harrell's c-index was calculated to compare the performance of each liver fibrosis biomarker. Statistical analyses were performed using Stata 15.0 (StataCorp, College Station, TX, USA). P < 0.05 was considered to be significant.

## Results

Table 1 shows the subjects' baseline characteristics. Of the 7,702 subjects, 1,141 (14.8%) died. The average duration of follow-up was 10.3 ± 2.2 years. In non-deceased subjects, the follow-

**Table 1. Baseline characteristics of the subjects.**

| | Total | Non-deceased | Deceased | P value |
|---|---|---|---|---|
| N (%) | 7,702 (100.0) | 6,561 (85.2) | 1,141 (14.8) | |
| Follow-up duration (years) | 10.3±2.2 | 10.9±1.0 | 6.4±3.1 | <0.001 |
| Male (%) | 2,595 (33.7) | 2,017 (30.7) | 578 (50.7) | <0.001 |
| Age (years) | 65.4±8.2 | 64.1±7.5 | 72.8±8.0 | <0.001 |
| Height (cm) | 157.4±8.2 | 157.4±7.9 | 157.4±9.5 | 0.896 |
| Weight (kg) | 60.5±9.1 | 60.8±8.9 | 58.6±10.2 | <0.001 |
| BMI (kg/m$^2$) | 24.4±2.9 | 24.5±2.9 | 23.6±3.1 | <0.001 |
| Waist circumference (cm) | 88.0±8.6 | 88.0±8.5 | 87.6±9.5 | 0.088 |
| Smoking (%) | 649 (8.4) | 507 (7.7) | 142 (12.4) | <0.001 |
| Alcohol intake (%) | 3,944 (51.2) | 3,368 (51.3) | 576 (50.5) | <0.001 |
| [a]Regular walking (%) | 4,809 (62.4) | 4,160 (63.4) | 649 (56.9) | <0.001 |
| [b]Hypertension (%) | 3,400 (44.2) | 2,779 (42.4) | 621 (54.4) | <0.001 |
| [c]Diabetes (%) | 1,609 (20.9) | 1,239 (18.9) | 370 (32.4) | <0.001 |
| Systolic BP (mm Hg) | 122.9±16.8 | 122.2±16.6 | 127.2±17.3 | <0.001 |
| HbA$_{1c}$ (%) | 5.8±0.9 | 5.8±0.9 | 6.1±1.1 | <0.001 |
| Total cholesterol (mg/dL) | 202.7±39.9 | 204.0±39.2 | 195.7±42.9 | <0.001 |
| Triglycerides (mg/dL) | 139.6±87.6 | 140.0±89.0 | 137.4±79.3 | 0.352 |
| HDL cholesterol (mg/dL) | 51.2±11.7 | 51.5±11.6 | 49.6±12.1 | <0.001 |
| AST (IU/L) | 24.1±15.5 | 23.8±13.4 | 25.6±24.0 | <0.001 |
| ALT (IU/L) | 20.0±13.9 | 20.0±13.1 | 20.0±17.8 | 0.932 |
| GGT (IU/L) | 29.6±49.9 | 28.2±34.0 | 37.6±100.4 | <0.001 |
| Platelet (IU/L) | 251.5±62.2 | 252.1±60.2 | 248.0±72.2 | 0.037 |
| Albumin (IU/L) | 4.5±0.3 | 4.5±0.2 | 4.4±0.3 | <0.001 |
| NFS | -1.6±1.2 | -1.7±1.1 | -1.1±1.3 | <0.001 |
| FIB-4 | 1.6±1.0 | 1.5±0.9 | 1.9±1.5 | <0.001 |
| APRI | 0.3±0.3 | 0.3±0.2 | 0.3±0.4 | <0.001 |
| BARD score | 2.2±0.7 | 2.2±0.7 | 2.3±0.7 | <0.001 |

All values are given as $n$ (%) or mean ± standard deviation.

BMI, body mass index; BP, blood pressure; HbA$_{1c}$, glycated hemoglobin; HDL, high-density lipoprotein; AST, aspartate aminotransferase; ALT, alanine aminotransferase; GGT, Gammaglutamyl transpeptidase; NFS, NAFLD fibrosis score; FIB-4, Fibrosis-4 score; APRI, aspartate aminotransferase to platelet ratio index.

[a]Subjects who performed walking for more than 30 minutes at one time and more than 5 times per week.

[b]Hypertension was defined as systolic blood pressure ≥140mm Hg or diastolic blood pressure ≥90mm Hg or taking anti-hypertension medication.

[c]Diabetes was defined as fasting serum glucose ≥126 mg/dl or taking insulin or oral diabetes medication.

up duration, weight, BMI, alcohol intake, regular walking, total cholesterol concentration, HDL cholesterol concentration, platelet concentration, and albumin concentration were higher; in deceased subjects, the frequency of male sex, older age, history of smoking, HTN, DM, SBP, HbA$_{1c}$ concentration, AST concentration, GGT concentration, NFS, FIB-4, APRI, and BARD score was higher.

The characteristics of the subjects according to the level of NFS, FIB-4, APRI, and BARD score are shown in Table 2. The follow-up duration, sex distribution, height, HTN prevalence, triglycerides concentration, AST concentration, ALT concentration, GGT concentration, and platelet concentration differed significantly according to the level of NFS, FIB-4, APRI, and BARD score. In addition, albumin concentration decreased significantly with increasing level of NFS, FIB-4, APRI, and BARD score; age increased significantly with increasing level of NFS, FIB-4, APRI, and BARD score. The deceased by overall increased significantly as NFS,

**Table 2. Characteristics of the subjects according to the level of NFS, FIB-4, APRI, and BARD score.**

| Variable | NFS | | | FIB-4 | | | APRI | | | BARD score | | |
|---|---|---|---|---|---|---|---|---|---|---|---|---|
| | Low risk (<-1.455) | Intermediate risk (-1.455–0.676) | High risk (>0.676) | Low risk (<1.30) | Intermediate risk (1.30–2.67) | High risk (>2.67) | Low risk (<0.5) | Intermediate risk (0.5–1.5) | High risk (>1.5) | Low risk (0–1) | Intermediate risk (2–3) | High risk (4) |
| N (%) | 4,444 (57.7) | 3,058 (39.7) | 200 (2.6) | 3,175 (41.2) | 4,090 (53.1) | 437 (5.7) | 7,245 (94.1) | 420 (5.5) | 37 (0.5) | 489 (6.3) | 7,017 (91.1) | 196 (2.5) |
| Follow-up duration (years) | 10.6±1.9 | 9.9±2.4 | 8.2±3.3* | 10.6 ±1.9 | 10.1±2.2 | 8.9±3.1* | 10.3 ±2.2 | 9.8±2.6 | 8.0 ±3.6* | 10.9 ±2.2 | 10.2±2.2 | 10.3 ±1.8* |
| Male (%) | 1,225 (27.6) | 1,280 (41.9) | 90 (45.0)* | 812 (25.6) | 1,561 (38.2) | 222 (50.8)* | 2,401 (33.1) | 178 (42.4) | 16 (43.2)* | 224 (45.8) | 2,327 (33.2) | 44 (22.4)* |
| Age (years) | 62.8±7.5 | 68.6±7.7 | 73.3 ±8.0* | 61.7 ±7.2 | 67.5±7.7 | 72.2 ±8.1* | 65.3 ±8.2 | 66.8±8.4 | 67.5 ±6.9* | 63.0 ±7.2 | 65.5±8.3 | 66.6 ±7.8* |
| Height (cm) | 157.0±7.8 | 157.9±8.5 | 157.1 ±10.0* | 156.8 ±7.7 | 157.7±8.4 | 158.5 ±9.3* | 157.3 ±8.2 | 158.7±8.3 | 158.2 ±9.9** | 159.9 ±8.0 | 157.3±8.2 | 155.3 ±7.9* |
| Weight (kg) | 59.3±8.6 | 62.0±9.5 | 62.4 ±10.7* | 60.6 ±8.7 | 60.4±9.3 | 60.0 ±10.0 | 60.3 ±9.1 | 62.9±9.8 | 61.2 ±12.3* | 64.2 ±9.3 | 59.9±8.9 | 72.0 ±8.2* |
| BMI (kg/m$^2$) | 24.0±2.7 | 24.8±3.1 | 25.2 ±3.3* | 24.6 ±2.8 | 24.3±3.0 | 23.8 ±3.2* | 24.3 ±2.9 | 24.9±3.4 | 24.3 ±3.2* | 25.1 ±2.6 | 24.2±2.8 | 29.8 ±1.7* |
| Waist circumference (cm) | 87.0±8.3 | 89.2±8.9 | 90.2 ±9.0* | 88.5 ±8.2 | 87.7±8.8 | 86.9 ±9.7* | 87.9 ±8.5 | 89.5±9.7 | 87.8 ±9.4** | 90.9 ±7.3 | 87.4±8.5 | 100.1 ±6.4* |
| Smoking (%) | 372 (8.4) | 261 (8.5) | 16 (8.0)* | 280 (8.8) | 335 (8.2) | 34 (7.8)* | 606 (8.4) | 38 (9.0) | 5 (13.5) | 53 (10.8) | 577 (8.2) | 19 (9.7)* |
| Alcohol intake (%) | 2,292 (51.6) | 1,537 (50.3) | 115 (57.5)** | 1,600 (50.4) | 2,112 (51.6) | 232 (53.1) | 3,716 (51.3) | 207 (49.3) | 21 (56.8) | 210 (42.9) | 3,620 (51.6) | 114 (58.2)** |
| [a]Regular walking (%) | 2,806 (63.1) | 1,885 (61.6) | 118 (59.0) | 1,950 (61.4) | 2,593 (63.4) | 266 (60.9) | 4,516 (62.3) | 269 (64.0) | 24 (64.9) | 312 (63.8) | 4,393 (62.6) | 104 (53.1)** |
| [b]Hypertension (%) | 1,722 (38.8) | 1,555 (50.9) | 123 (61.5)* | 1,287 (40.5) | 1,882 (46.0) | 231 (52.9)* | 3,154 (43.5) | 225 (53.6) | 21 (56.8)* | 231 (47.2) | 3,032 (43.2) | 137 (69.9)* |
| [c]Diabetes (%) | 389 (8.8) | 1,095 (35.8) | 125 (62.5)* | 699 (22.0) | 815 (19.9) | 95 (21.7) | 1,478 (20.4) | 115 (27.4) | 16 (43.2)* | 185 (37.8) | 1,228 (17.5) | 196 (100.0)* |
| Systolic BP (mm Hg) | 121.6 ±16.7 | 124.6±16.7 | 126.6 ±18.1* | 121.6 ±16.6 | 123.6±16.9 | 125.8 ±17.0* | 122.8 ±16.8 | 124.3±16.9 | 125.1 ±15.8 | 123.0 ±14.7 | 122.8±17.0 | 126.7 ±16.4** |
| HbA$_{1c}$ (%) | 5.7±0.8 | 6.1±1.1 | 6.2±1.1* | 5.9±1.0 | 5.8±0.9 | 5.7±0.8* | 5.8 ±0.9 | 5.9±1.0 | 5.7±1.0 | 6.3 ±1.4 | 5.8±0.9 | 7.0±1.1* |
| Total cholesterol (mg/dL) | 207.5 ±38.7 | 196.7±40.3 | 190.0 ±43.7* | 206.6 ±40.1 | 201.0±39.2 | 190.7 ±41.5* | 203.2 ±39.5 | 197.1±44.6 | 181.3 ±48.5* | 202.9 ±44.6 | 202.9±39.5 | 197.1 ±42.8 |
| Triglycerides (mg/dL) | 143.9 ±92.0 | 134.4±80.9 | 125.2 ±81.6* | 149.7 ±93.1 | 134.0±83.5 | 119.3 ±74.7* | 139.1 ±85.4 | 149.9±118.2 | 118.2 ±95.4** | 167.1 ±107.7 | 136.9±85.4 | 168.6 ±93.0* |
| HDL cholesterol (mg/dL) | 51.8±11.7 | 50.4±11.7 | 49.5 ±12.9* | 50.9 ±11.4 | 51.4±11.8 | 51.5 ±12.4 | 51.2 ±11.6 | 51.0±13.4 | 51.9 ±16.5 | 49.8 ±11.7 | 51.4±11.7 | 47.9 ±10.1* |
| AST (IU/L) | 23.1±13.0 | 24.8±16.7 | 33.8 ±32.7* | 20.9 ±5.2 | 24.6±7.6 | 41.8 ±55.6* | 22.4 ±5.4 | 41.4±15.2 | 146.6 ±151.9* | 26.3 ±14.8 | 23.8±15.5 | 28.4 ±14.3* |
| ALT (IU/L) | 20.7±13.1 | 19.2±14.8 | 17.1 ±16.3* | 20.2 ±9.6 | 19.2±10.9 | 26.2 ±39.7* | 18.6 ±8.1 | 37.8±23.9 | 101.9 ±103.2* | 39.1 ±23.3 | 18.5±11.8 | 25.0 ±15.2* |
| GGT (IU/L) | 27.8±33.0 | 29.4±34.9 | 71.3 ±226.8* | 27.4 ±32.4 | 28.1±28.7 | 58.9 ±166.4* | 26.9 ±28.5 | 54.3±62.6 | 276.1 ±503.0* | 44.2 ±65.7 | 28.3±48.9 | 37.6 ±26.9* |
| Platelet (IU/L) | 278.5 ±57.9 | 217.6±44.9 | 169.3 ±56.2* | 291.1 ±58.4 | 230.1±43.7 | 164.2 ±46.6* | 256.2 ±59.5 | 180.5±51.8 | 132.0 ±72.7* | 246.1 ±60.3 | 251.8±62.1 | 255.6 ±69.3* |
| Albumin (IU/L) | 4.5±0.2 | 4.4±0.3 | 4.3±0.4* | 4.5±0.3 | 4.5±0.3 | 4.4±0.4* | 4.5 ±0.3 | 4.5±0.3 | 4.0 ±0.6* | 4.6 ±0.3 | 4.5±0.3 | 4.4±0.2* |
| NFS | -2.4±0.7 | -0.7±0.5 | 1.4±0.8* | -2.5±0.9 | -1.2±0.8 | 0.4±1.0* | -1.7 ±1.1 | -0.5±1.2 | 0.9 ±1.8* | -2.1 ±1.0 | -1.6±1.1 | -0.3 ±1.2* |

*(Continued)*

**Table 2.** (Continued)

| Variable | NFS | | | FIB-4 | | | APRI | | | BARD score | | |
|---|---|---|---|---|---|---|---|---|---|---|---|---|
| | Low risk (<-1.455) | Intermediate risk (-1.455–0.676) | High risk (>0.676) | Low risk (<1.30) | Intermediate risk (1.30–2.67) | High risk (>2.67) | Low risk (<0.5) | Intermediate risk (0.5–1.5) | High risk (>1.5) | Low risk (0–1) | Intermediate risk (2–3) | High risk (4) |
| FIB-4 | 1.2±0.3 | 1.9±0.6 | 4.4±4.2* | 1.0±0.2 | 1.7±0.3 | 3.9±2.9* | 1.4±0.5 | 2.8±1.2 | 9.8±7.4* | 1.1±0.5 | 1.6±1.0 | 1.8±2.2* |
| APRI | 0.3±0.2 | 0.3±0.2 | 0.8±1.0* | 0.2±0.1 | 0.3±0.1 | 0.8±1.0* | 0.3±0.1 | 0.7±0.2 | 3.2±2.1* | 0.3±0.3 | 0.3±0.3 | 0.4±0.5* |
| BARD score | 2.0±0.6 | 2.4±0.7 | 2.9±0.7* | 2.1±0.8 | 2.2±0.6 | 2.3±0.6* | 2.2±0.7 | 2.2±0.9 | 2.4±0.9 | 0.5±0.5 | 2.2±0.4 | 4.0±0.0* |
| Deceased by all-cause (%) | 434 (9.8) | 626 (20.5) | 81 (40.5)* | 307 (9.7) | 689 (16.8) | 145 (33.2)* | 1,033 (14.3) | 91 (21.7) | 17 (45.9)* | 62 (12.7) | 1,052 (15.0) | 27 (13.8) |
| Deceased by CVD (%) | 13 (0.3) | 34 (1.1) | 4 (2.0)* | 11 (0.3) | 32 (0.8) | 8 (1.8)* | 50 (0.7) | 1 (0.2) | 0 (0.0) | 4 (0.8) | 47 (0.7) | 0 (0.0) |

All values are given as *n* (%) or mean ± standard deviation.

*p<0.001,

**p<0.05.

NFS, NAFLD fibrosis score; BMI, body mass index; BP, blood pressure; HbA$_{1c}$, glycated hemoglobin; HDL, high-density lipoprotein; AST, aspartate aminotransferase; ALT, alanine aminotransferase; GGT, gamma-glutamyl transpeptidase; FIB-4, Fibrosis-4 score; APRI, aspartate aminotransferase to platelet ratio index.

[a]Subjects who performed walking for more than 30 minutes at one time and more than 5 times per week.

[b]Hypertension was defined as systolic blood pressure ≥140mm Hg or diastolic blood pressure ≥90mm Hg or taking anti-hypertension medication.

[c]Diabetes was defined as fasting serum glucose ≥126 mg/dl or taking insulin or oral diabetes medication.

FIB-4, and APRI levels increased and the deceased by CVD increased significantly as NFS and FIB-4 levels increased.

The hazard ratios (HRs) for overall and CVD mortality according to the level of NFS, FIB-4, APRI, and BARD score are shown in Table 3. With the multivariate adjustment, overall mortality increased with increasing NFS level [HR 1.9, 95% confidence interval (CI) 1.7–2.2 for intermediate risk vs. low risk; HR 4.3, 95% CI 3.3–5.5 for high risk vs. low risk], increasing FIB-4

**Table 3. Hazard ratios for overall and CVD mortality according to the level of NFS, FIB-4, APRI, and BARD score.**

| Variables | | NFS | | FIB-4 | | APRI | | BARD score | |
|---|---|---|---|---|---|---|---|---|---|
| | | Non-adjusted | Adjusted[a] | Non-adjusted | Adjusted[b] | Non-adjusted | Adjusted[c] | Non-adjusted | Adjusted[d] |
| Overall mortality | Low risk | Reference | Reference | Reference | Reference | Reference | Reference | Reference | Reference |
| | Intermediate risk | 2.3 (2.0–2.6) | 1.9 (1.7–2.2) | 1.9 (1.6–2.1) | 1.7 (1.5–2.0) | 1.6 (1.3–2.0) | 1.3 (1.0–1.6) | 1.3 (1.0–1.7) | 1.2 (0.9–1.6) |
| | High risk | 5.8 (4.5–7.3) | 4.3 (3.3–5.5) | 4.3 (3.6–5.3) | 3.5 (2.9–4.4) | 4.4 (2.7–7.1) | 3.5 (2.1–5.8) | 1.2 (0.8–1.9) | 0.9 (0.6–1.4) |
| CVD mortality | Low risk | Reference | Reference | Reference | Reference | Reference | Reference | Reference | Reference |
| | Intermediate & high risk | 4.3 (2.3–8.2) | 3.6 (1.8–6.9) | 2.7 (1.4–5.3) | 2.6 (1.3–5.2) | 0.3 (0.5–2.5) | 0.3 (0.1–1.9) | 1.2 (0.4–3.3) | 0.5 (0.1–1.5) |

NFS, NAFLD fibrosis score; FIB-4, Fibrosis-4 score; APRI, aspartate aminotransferase to platelet ratio index.

[a]adjusted by sex, log-waist circumference, smoking, alcohol intake, regular walking, hypertension, log-SBP, log-HbA$_{1c}$, log-total cholesterol, log-triglyceride, log-HDL, and log-GGT.

[b]adjusted by sex, log-waist circumference, smoking, alcohol intake, regular walking, hypertension, diabetes, log-SBP, log-HbA$_{1c}$, log-total cholesterol, log-triglyceride, log-HDL, and log-GGT.

[c]adjusted by sex, age, log-waist circumference, smoking, alcohol intake, regular walking, hypertension, diabetes, log-SBP, log-HbA$_{1c}$, log-total cholesterol, log-triglyceride, log-HDL, and log-GGT.

[d]adjusted by sex, age, log-waist circumference, smoking, alcohol intake, regular walking, hypertension, log-SBP, log-HbA$_{1c}$, log-total cholesterol, log-triglyceride, log-HDL, and log-GGT.

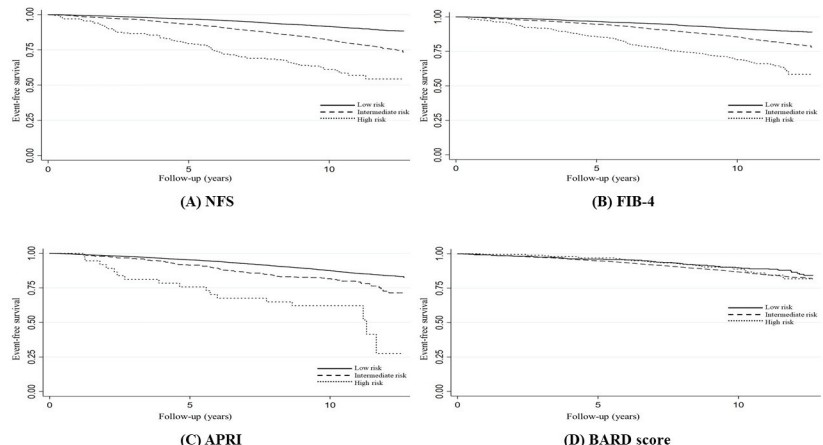

**Fig 1. Kaplan-Meier survival estimates between NFS, FIB-4, APRI, and BARD score for overall mortality.**

level (HR 1.7, 95% CI 1.5–2.0 for intermediate risk vs. low risk; HR 3.5, 95% CI 2.9–4.4 for high risk vs. low risk), and increasing APRI level (HR 1.3, 95% CI 1.0–1.6 for intermediate risk vs. low risk; HR 3.5, 95% CI 2.1–5.8 for high risk vs. low risk) but not with BARD score. Additionally, with multivariate adjustment, CVD mortality increased with increasing NFS level (HR 3.6, 95% CI 1.8–6.9 for intermediate/high risk vs. low risk), increasing FIB-4 level (HR 2.6, 95% CI 1.3–5.2 for intermediate/high risk vs. low risk) but not with APRI and BARD score.

In Kaplan-Meier survival estimates (Fig 1), overall survival was lower at higher risk for NFS, FIB-4, and APRI (Log-rank test p<0.001), but not for BARD score (Log-rank test p = 0.126).

Table 4 shows the comparison of Harrell's c-index between NFS, FIB-4, APRI, and BARD score. The Harrell's c-index (95% CI) showed that the NFS and FIB-4 were more predictive of overall mortality, with 0.643 (0.626–0.660) and 0.633 (0.616–0.650), compared with that of 0.508 (0.490–0.527) for APRI and 0.548 (0.533–0.562) for BARD score.

**Table 4. Comparison of the Harrell's c-index between NFS, FIB-4, APRI, and BARD score for the discrimination of overall mortality.**

|  | Harrell's c-index (95% CI) | *P value* |
|---|---|---|
| **Biomarkers** | | |
| NFS | 0.643 (0.660–0.626) | |
| FIB-4 | 0.633 (0.650–0.616) | |
| APRI | 0.508 (0.527–0.490) | |
| BARD score | 0.548 (0.562–0.533) | |
| **Differences between biomarkers** | | |
| NFS-FIB-4 | 0.010 (0.022 –-0.001) | 0.507 |
| NFS-APRI | 0.135 (0.153–0.117) | <0.001 |
| NFS-BARD score | 0.095 (0.114–0.077) | <0.001 |
| FIB-4-APRI | 0.125 (0.137–0.112) | <0.001 |
| FIB-4-BARD score | 0.085 (0.107–0.063) | <0.001 |
| APRI-BARD score | -0.039 (-0.016 –-0.063) | 0.007 |

c-index, concordance index; CI, confidence interval; NFS, NAFLD fibrosis score; FIB-4, Fibrosis-4 score; APRI, aspartate aminotransferase to platelet ratio index.

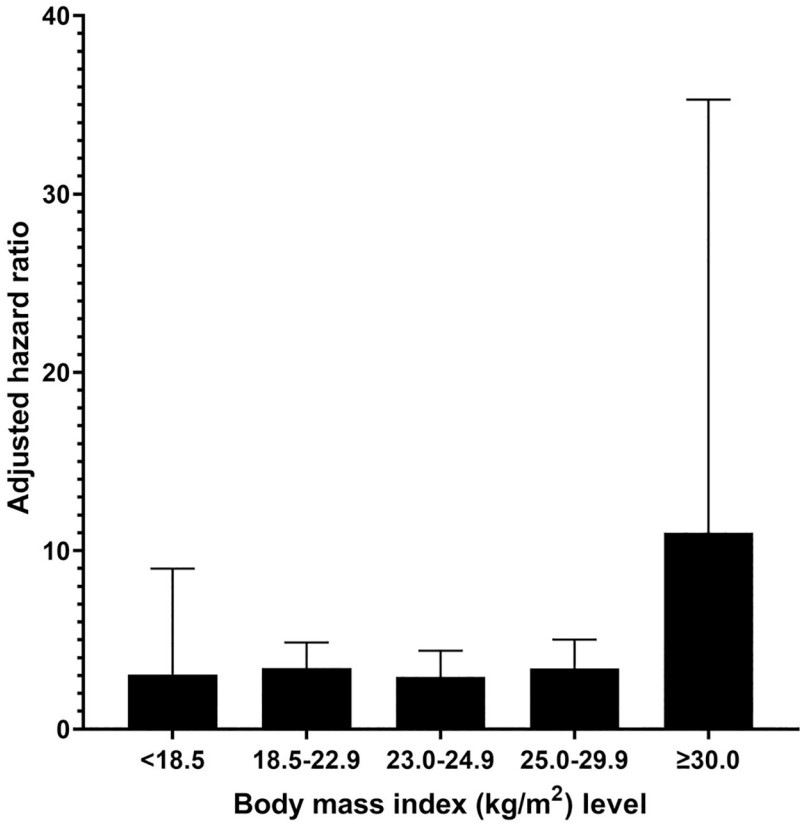

**Fig 2. Adjusted hazard ratios of FIB-4 for overall mortality according to the BMI level.**

The adjusted HRs of FIB-4 for overall mortality according to the BMI level are shown in Fig 2. Comparing high risk level with low risk level of FIB-4, the adjusted HRs for higher BMI were higher than those for lower BMI.

## Discussion

We examined the links between liver fibrosis biomarkers and overall CVD mortality in the Korean population aged ≥50 years. After controlling for covariates, NFS, FIB-4, and APRI showed a significant relationship with the overall mortality, and NFS and FIB-4 showed a significant relationship with the CVD mortality. Furthermore, the Harrell's c-index revealed that the NFS and FIB-4 were more predictive of overall mortality than the APRI and BARD score.

In most previous studies, liver fibrosis biomarkers showed a consistent association with overall and liver-related mortality [8], but the relationship between liver fibrosis and CVD mortality was inconsistent. In two studies using NHANES [20,21], liver fibrosis as measured by NFS and FIB-4 was significantly associated with CVD mortality. Similarly, in another study with the Italian cohort aged 65 years or older, NFS and FIB-4 showed a significant association with CVD mortality [22]. However, in a cohort study with Brazilian patients with diabetes, the authors demonstrated that NFS and FIB-4 were significantly associated with overall mortality, but not with CVD mortality [7]. Additionally, in a 16-year prospective cohort study with the Korean general population [23], the NFS and FIB-4 were significantly associated with overall and liver-specific mortality, but not with CVD mortality, which was different from our results. This may be due to variations in the study population, follow-up periods, and measures of

liver fibrosis. In particular, the prior Korean study set cutoff values of NFS and FIB-4 to −2.08 and 1.22, respectively [23]. This may have lowered the strength of the association by including subjects with early liver fibrosis. In a previous cohort study with patients with NAFLD, only those with fibrosis 3–4 had increased CVD mortality [24].

Liver fibrosis is developed by a variety of etiologies, such as hepatitis B and C virus infections, metabolic disorders, and excessive alcohol intake [25]. Fibrosis is a long-term, staggering wound healing process that results increased deposition of extracellular matrix [26]. Persistent fibrosis causes cirrhosis and stiffness of the liver, impairing its physiological function, which can eventually lead to cirrhosis of the liver, end-stage liver disease, or hepatocellular cancer [25,27]. In a multicenter cohort study of 320 NAFLD patients, higher liver fibrosis is associated with increased risk for liver related complications such as gastroesophageal bleeding, hepatopulmonary syndrome, cirrhosis complication, hepatocellular carcinoma, and liver transplantation [9]. Moreover, other epidemiologic studies have indicated that liver fibrosis is associated with aortic stiffness [28], heart failure, atrial fibrillation, and coronary heart disease [29–32].

Liver biopsy is the best method for risk stratification of patients with NAFLD, but due to some limitations, various simple biomarkers have been developed. Therefore, it has been very important to find the most accurate predictor among various biomarkers. In several systematic reviews [4,8,10], investigators analyzed massive cohort data and demonstrated that FIB-4 and NFS showed higher prognostic accuracy for the mortality than APRI. Accordingly, AASLD guidelines and EASL endorsed NFS and FIB-4 as a diagnostic measure for ruling out advanced liver fibrosis [11,12]. However, there are limitations in applying the optimal biomarker to Korean patients with NAFLD due to the lack of studies comparing the performance of liver fibrosis biomarkers with Korean people. In the present study, the Harrell's c-index revealed that the NFS and FIB-4 were more predictive of overall mortality than the APRI and BARD score, similar to previous studies. In this study, BARD score and APRI showed lower performance than NFS and FIB-4, but BARD score and APRI also have advantages. The BARD score can predict histological fibrosis with reasonable accuracy. Researchers reported that area under curve ROC of BARD score was 0.67 in predicting fibrosis, which was similar to 0.68 of NFS [6]. In addition, since APRI can be easily calculated with only two indicators (AST and platelets), the World Health Organization (WHO) recommends APRI to determine the stage of fibrosis in countries with limited resources [33].

In the present study, the HRs for overall mortality according to the BMI level were higher in obese subjects than in lean subjects. Several previous studies reported an "obesity paradox" in which the mortality rate of obese subjects with CLD was lower than that of subjects with normal weight [34–36]. However, the evidence for the obesity paradox in CLD was not robust. In the study with 1999–2016 NHANES data [37], the 15-year cumulative overall mortality was higher in nonobese patients with NAFLD than in obese patients with NAFLD. However, after adjusting for other covariates, the association between nonobese NAFLD and high overall mortality disappeared. Additionally, in a Swedish cohort study of 646 patients with biopsy-proven NAFLD [38], lean patients with NAFLD had an increased risk of developing serious liver disease, but no association with the overall mortality rate. In the analysis with the Korean cohort study [23], the authors demonstrated that the association between liver fibrosis and mortality was more prominent in lean subjects than in obese subjects, which was inconsistent with our results. However, the obesity paradox requires some careful attention in interpreting the results [39]. First, there is a limitation of BMI as an obesity indicator. Second, patients with severe CLD lose weight more. Third, there might be a survivor bias that more obese patients have already died.

This is a well-established, large general population cohort study with an extremely high follow-up rate. Furthermore, to the best of our knowledge, this is the first study to compare the

performance of liver fibrosis biomarkers in a Korean population. However, this study has some limitations. First, accurate information about liver fibrosis was lacking due to the general population's lack of access to liver biopsy. Second, data on the severity of hypertension and diabetes were not available. Instead, SBP and HbA$_{1c}$ were used as surrogates to adjust for blood pressure and diabetes severity. Third, since all potential confounding factors, such as progression and reversal of liver fibrosis, and changes in body mass index (BMI) and NAFLD, were not adjusted for, residual confounding may affect the main results. Finally, information on drugs for hypertension, diabetes, and hyperlipidemia has not been measured. Recently, some studies reported that the use of statin [40], angiotensin converting enzyme inhibitor (ACEI) [41], and diabetic medication [42] reduce the risk of advanced fibrosis.

## Conclusion

The NFS, FIB-4, and APRI showed a significant relationship with the overall mortality, and NFS and FIB-4 showed a significant relationship with the CVD mortality. In addition, the Harrell's c-index showed that the NFS and FIB-4 were more predictive of overall mortality than the APRI and BARD score in Korean adults aged ≥50 years.

## Author Contributions

**Conceptualization:** Seong-Woo Choi, Min-Ho Shin.

**Data curation:** Young-Hoon Lee, So-Yeon Ryu.

**Formal analysis:** Sun-Seog Kweon.

**Investigation:** Seong-Woo Choi, Sun-Seog Kweon, Hae-Sung Nam.

**Methodology:** Sun-Seog Kweon, So-Yeon Ryu.

**Supervision:** Min-Ho Shin.

**Validation:** Min-Ho Shin.

**Visualization:** Young-Hoon Lee.

**Writing – original draft:** Seong-Woo Choi.

**Writing – review & editing:** Hae-Sung Nam, Min-Ho Shin.

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
