## [Decision Letter · Decision Letter 0]

26 Aug 2022

PONE-D-22-22156Association of liver fibrosis biomarkers with overall and CVD mortality in the Korean population: The Dong-gu StudyPLOS ONE

Dear Dr. Choi,

Thank you for submitting your manuscript to PLOS ONE. After careful consideration, we feel that it has merit but does not fully meet PLOS ONE’s publication criteria as it currently stands. Therefore, we invite you to submit a revised version of the manuscript that addresses the points raised during the review process.

We look forward to receiving your revised manuscript.

Kind regards,

Taeyun Kim

Academic Editor

PLOS ONE

Journal Requirements:

Additional Editor Comments:

* Please format the title page, manuscript, and references according to PLOS ONE guidelines.

* In tables, when expressing the confidence interval, please use en dash. The tilde is not universally used for expressing the range. Also please correct other type errors.

* It would be more informative to draw Kaplan-Meier curve to compare the probability of death between low, intermediate, and high risk.

* Ascertainment of death-"Linking with National Statistical Office registries yielded death information. Until December". What this sentence means? Only death event could be confirmed? Not date?

* ICD-10 I20-25 cover cardiovascular disease. But, I60-69 cover "cerebrovascular disease". Although they share same risk factors, cardiovascular disease is a disease of the blood vessels in the heart, and cerebrovascular disease is a disease of the blood vessels in the brain. Therefore, it seems irresolute to combine them into cardiovascular disease.

* Moreover, I wonder the accuracy of the disease-specific death. For example, ICD-10 I20 is angina pectoris which could be not severe and I25 is a kind of chronic disease. Then, how the authors could be confident on their definition on cardiovascular death? Isn't there another possible main cause of death such as malignancy?

* Please discuss the possible mechanisms and experimental studies around the negative effect of liver fibrosis on the clinical outcomes. 

* Several additional limitations have been raised by Reviewers. Please separate the paragraph to individually discuss the study limitations. And, please be more specific and clear, not to be ambiguous on this section.

Reviewers' comments:

Reviewer's Responses to Questions

**Comments to the Author**

1. Is the manuscript technically sound, and do the data support the conclusions?

Reviewer #1: Yes

Reviewer #2: No

Reviewer #3: Yes

2. Has the statistical analysis been performed appropriately and rigorously? 

Reviewer #1: Yes

Reviewer #2: No

Reviewer #3: I Don't Know

3. Have the authors made all data underlying the findings in their manuscript fully available?

Reviewer #1: No

Reviewer #2: No

Reviewer #3: Yes

4. Is the manuscript presented in an intelligible fashion and written in standard English?

Reviewer #1: Yes

Reviewer #2: Yes

Reviewer #3: Yes

5. Review Comments to the Author

Reviewer #1: This Is a well-written study based on a large population sample with a long follow-up showing that liver fibrosis biomarkers are associated to overall and cardiovascular mortality.

Results from this study confirm notions from previous studies showing that non invasive liver fibrosis indexes were correlated with cardiovascular risk and events (e.g. Pisetta, C.; Chillè, C.; Pelizzari, G.; Pigozzi, M.G.; Salvetti, M.; Paini, A.; Muiesan, M.L.; De Ciuceis, C.; Ricci, C.; Rizzoni, D. Evaluation of Cardiovascular Risk in Patient with Primary Non-alcoholic Fatty Liver Disease. High Blood Press. Cardiovasc. Prev. 2020, 27, 321–330.

Ballestri S, Mantovani A, Baldelli E, Lugari S, Maurantonio M, Nascimbeni F, Marrazzo A, Romagnoli D, Targher G, Lonardo A. Liver Fibrosis Biomarkers Accurately Exclude Advanced Fibrosis and Are Associated with Higher Cardiovascular Risk Scores in Patients with NAFLD or Viral Chronic Liver Disease. Diagnostics (Basel). 2021 Jan 9;11(1):98. doi: 10.3390/diagnostics11010098. PMID: 33435415; PMCID: PMC7827076.

Baratta, F.; Pastori, D.; Angelico, F.; Balla, A.; Paganini, A.M.; Cocomello, N.; Ferro, D.; Violi, F.; Sanyal, A.J.; Del Ben, M. Nonalcoholic Fatty Liver Disease and Fibrosis Associated With Increased Risk of Cardiovascular Events in a Prospective Study. Clin. Gastroenterol. Hepatol. 2020, 18, 2324–2331.e4).

NAFLD is an epidemic thrombofilic condition thus many patients with advance liver disease will suffer from cardiovascular disease and will be candidate to treatment (Spinosa, M.; Stine, J.G. Nonalcoholic Fatty Liver Disease-Evidence for a Thrombophilic State? Curr. Pharm. Des. 2020, 26, 1036–1044. Ballestri, S.; Capitelli, M.; Fontana, M.C.; Arioli, D.; Romagnoli, E.; Graziosi, C.; Lonardo, A.; Marietta, M.; Dentali, F.; Cioni, G. Direct Oral Anticoagulants in Patients with Liver Disease in the Era of Non-Alcoholic Fatty Liver Disease Global Epidemic: A Narrative Review. Adv. Ther. 2020, 37, 1910–1932.).

Please discuss and add appropriate literature references.

Reviewer #2: This study aimed to assess associations of NFS, FIB-4, APRI and BARD scores with mortality using a large cohorts 7,702 subjects. The main results were that NFS, FIB-4, and APRI were significantly associated with overall and CVD mortality with median duration of follow-up 10.3 years.

Basically, it is quite difficult to evaluate associations of noninvasive fibrosis markers with mortality in the longitudinal cohort because lots of potential, but important confounders cannot be controlled such as change of BMI, NAFLD, even fibrosis reversal and progression during time course of about 10 years in this study. Furthermore, regarding CVD mortality, exposure to statin use and control of diabetes (new onset during periods) were not considered in the Cox analyses.

Reviewer #3: Seong-Woo Choi et al evaluated the associations of liver fibrosis biomarkers [non-alcoholic fatty liver disease fibrosis score (NFS), fibrosis-4 (FIB-4), aspartate aminotransferase/platelet ratio index (APRI),

and BARD score] with mortality in Korean adults aged ≥50 years. Furthermore, the Harrell's c-index

revealed that the NFS and FIB-4 were more predictive of overall and CVD mortality than the APRI and BARD

score.The NFS, FIB-4, and APRI were significantly associated with overall and CVD mortality.This is a very interesting study with a long follow-up period and a large sample size. Although the relationship between NAFLD and CVD is certain. NAFLD usually has high BMI, abnormal lipid metabolism, hypertension, etc., the scientific value of this study is not have high significant,but it give the new methed to predict mortality.

There are a few minor issues:

1. 1. The severity of hypertension and diabetes in the cohort was not expanded in stratified analyses.

2, BARD score is not well explained in the article.

3, The impact of underlying diseases on mortality, such as the choice of therapeutic intervention and drug application, should be more addressed in the discussion.

6. PLOS authors have the option to publish the peer review history of their article (what does this mean?). If published, this will include your full peer review and any attached files.

Reviewer #1: No

Reviewer #2: No

Reviewer #3: No

---

## [Author Response · Author response to Decision Letter 0]

9 Oct 2022

I upload the file of response to reviewers.

---

## [Decision Letter · Decision Letter 1]

3 Nov 2022

Association of liver fibrosis biomarkers with overall and CVD mortality in the Korean population: The Dong-gu Study

PONE-D-22-22156R1

Dear Dr. Choi,

We’re pleased to inform you that your manuscript has been judged scientifically suitable for publication and will be formally accepted for publication once it meets all outstanding technical requirements.

Kind regards,

Taeyun Kim

Academic Editor

PLOS ONE

Additional Editor Comments (optional):

Reviewers' comments:

Reviewer's Responses to Questions

**Comments to the Author**

1. If the authors have adequately addressed your comments raised in a previous round of review and you feel that this manuscript is now acceptable for publication, you may indicate that here to bypass the “Comments to the Author” section, enter your conflict of interest statement in the “Confidential to Editor” section, and submit your "Accept" recommendation.

Reviewer #1: All comments have been addressed

Reviewer #3: All comments have been addressed

2. Is the manuscript technically sound, and do the data support the conclusions?

Reviewer #1: Yes

Reviewer #3: Yes

3. Has the statistical analysis been performed appropriately and rigorously? 

Reviewer #1: Yes

Reviewer #3: Yes

4. Have the authors made all data underlying the findings in their manuscript fully available?

Reviewer #1: Yes

Reviewer #3: Yes

5. Is the manuscript presented in an intelligible fashion and written in standard English?

Reviewer #1: Yes

Reviewer #3: Yes

6. Review Comments to the Author

Reviewer #1: The Authors have addressed all the reviewers' comments. The manuscript has substantially improved.

I feel that is suitable for publication.

Reviewer #3: OK,all comments have been addressed.Also fully discussed and explained for publication.no additional comments.

7. PLOS authors have the option to publish the peer review history of their article (what does this mean?). If published, this will include your full peer review and any attached files.

Reviewer #1: No

Reviewer #3: No

---

## [Editor Report · Acceptance letter]

5 Dec 2022

PONE-D-22-22156R1 

Association of liver fibrosis biomarkers with overall and CVD mortality in the Korean population: The Dong-gu Study 

Dear Dr. Choi:

I'm pleased to inform you that your manuscript has been deemed suitable for publication in PLOS ONE. Congratulations! Your manuscript is now with our production department. 

Kind regards, 

on behalf of

Dr. Taeyun Kim 

Academic Editor

PLOS ONE